# Reduced thermal expansion by surface-mounted nanoparticles in a pillared-layered metal-organic framework

Jan Berger [1], Alper-Sedat Dönmez[1], Aladin Ullrich [2], Hana Bunzen [2], Roland A. Fischer [1✉] & Gregor Kieslich [1✉]

Control of thermal expansion (TE) is important to improve material longevity in applications with repeated temperature changes or fluctuations. The TE behavior of metal-organic frameworks (MOFs) is increasingly well understood, while the impact of surface-mounted nanoparticles (NPs) on the TE properties of MOFs remains unexplored despite large promises of NP@MOF composites in catalysis and adsorbate diffusion control. Here we study the influence of surface-mounted platinum nanoparticles on the TE properties of Pt@MOF (Pt@$Zn_2$(DP-bdc)$_2$dabco; DP-bdc$^{2-}$=2,5-dipropoxy-1,4-benzenedicarboxylate, dabco=1,4-diazabicyclo[2.2.2]octane). We show that TE is largely retained at low platinum loadings, while high loading results in significantly reduced TE at higher temperatures compared to the pure MOF. These findings support the chemical intuition that surface-mounted particles restrict deformation of the MOF support and suggest that composite materials exhibit superior TE properties thereby excluding thermal stress as limiting factor for their potential application in temperature swing processes or catalysis.

[1] Chair of Inorganic and Metal-Organic Chemistry, TUM School of Natural Sciences, Technical University of Munich, Lichtenbergstr. 4, 85748 Garching, Germany. [2] Institute of Physics, University of Augsburg, Universitätsstr. 1, 86159 Augsburg, Germany. ✉email: roland.fischer@tum.de; gregor.kieslich@tum.de

The magnitude and sign of thermal expansion (TE) coefficients of a solid material is a key property that needs to be considered for a wide range of applications. Some major research goals related to the materials' thermal expansion behavior are the inhibition of TE to minimize stress from application-related repeating temperature changes in order to increase material longevity, i.e. of adsorbents during temperature swing adsorption or of supports and catalysts during process temperature fluctuations[1–4], the combination of different materials with positive (PTE) and negative (NTE) TE to reach near-zero thermal expansion (ZTE) across a certain temperature range to prevent tears, voids, or deformation[5–7], or the design of large (anisotropic) thermal expansion as applied in temperature-controlled sieves or sensors[8]. Naturally, these goals are of great importance for metal-organic frameworks (MOFs) and their potential applications. MOFs are a chemically diverse material class with guest accessible void space based on a modular building principle, providing a fascinating material platform for studying of how structural changes impact the macroscopic properties important for designing sensors, adsorbents, supports or catalysts amongst many other examples[9,10].

Looking at the thermal expansion behavior of MOFs, most MOFs exhibit NTE, a property that is still considered counterintuitive for purely inorganic solids[11–13]. Molecular simulations on mesoporous MOFs draw clear correlations between porosity, transverse ligand vibration and NTE, underpinning the role of available pore space as a NTE directing factor[14,15]. Approaches for tuning the TE behavior of MOFs encompasses all controllable variables in MOF design such as expansivity control by defect-engineering[16], linker functionalization[17–19], or changing metals ions or organic linker length[14]. Important examples are $Zn_2(bdc)_{2-x}(TM-bdc)_x dabco$ (TM-bdc$^{2-}$=tetramethyl-1,4-benzenedicarboxylate; dabco=1,4-diazabicyclo[2.2.2]octane) which exhibits a change from PTE to NTE as a function of bdc:TM-bdc ratio[20], and UiO-67 where linker functionalization leads to subtle changes in NTE behavior[21]. Additionally, certain network topologies can lead to pronounced anisotropic TE behavior and temperature-induced phase transitions, of which MOFs with a wine-rack type network such as MIL-53(Al), abovementioned $Zn_2(fu-bdc)_2 dabco$, and [Ag(en)]NO$_3$ (en=ethylenediamine) are prototypical examples[17,22–24]. These insights are relevant for future MOF applications, because even single-percentage temperature-induced framework contraction can lead to considerable softening and greatly impact mechanical stability[25]. In addition to framework modification, external factors affecting porosity such as interstitial guest molecules can similarly impact TE[26]. Examples are DUT-49 that switches from NTE in the guest-free state to PTE when solvated[27], or Prussian blue derivative frameworks showing a continuous NTE-to-PTE transition in response to CO$_2$ sorption[28]. The impact of nanoparticles as external factor influencing TE of composites consisting of nanoparticles integrated into or surface-mounted onto MOFs (NP@MOF) is yet unexplored, to the best of our knowledge. This is despite the large interest in these composite systems for catalysis[29–32], and the known importance of these properties for nanoparticles, classical supported catalyst composites or thin film composites[3,5,8,33].

Here we provide the first step in elucidating the influence of nanoparticles on the thermal expansion behavior in NP@MOF composites. We synthesize Pt@MOF composites by using a capping agent-free, one-pot co-formation of MOF and nanoparticle and investigate the effects of low Pt nanoparticle loading (1–3 wt.%) on the TE of the model pillared-layered MOF system $Zn_2(DP-bdc)_2 dabco$ (DP-bdc$^{2-}$=2,5-dipropoxy-1,4-benzenedicarboxylate) (see Fig. 1 and Supplementary Section S1.4)[17]. The results suggest that TE properties of composite systems can be estimated based on the TE properties of the MOF support, and that thermal stress in composite systems is expected to be smaller when compared to the parent MOF.

## Results and discussion

**Pt@MOF composite synthesis.** All Pt@Zn$_2$(DP-bdc)$_2$dabco composites were synthesized using a literature-inspired[34] co-crystallization approach which enables one-step one-pot crystallization of MOF and formation of Pt nanoparticles without the need for any capping agents. This method combines the MOF synthesis procedure, as known from literature[17], with a solvent mediated Pt$^{2+}$ reduction and provides control over Pt nanoparticle loading on the surface of the MOF support. MOF precursors Zn(NO$_3$)$_2$, dabco, H$_2$DPbdc and the Pt nanoparticle precursor K$_2$PtCl$_4$ are suspended in dimethylformamide (DMF). During solvothermal MOF synthesis, DMF oxidizes in presence of water (originating from trace amounts of H$_2$O in DMF and the hydrated metal salt) and Pt$^{2+}$ to form its carbamic acid derivative and reduced Pt$^0$, thereby facilitating nanoparticle formation[35]. Both MOF and nanoparticle form gradually and precipitate as composite. This circumvents the need for nanoparticle isolation and capping agents which is expected to improve reproducibility and correlation of nanoparticle impact and MOF behavior[36]. Modulation of reaction temperature, time and precursor concentration allows for some control over Pt nanoparticle formation kinetics and resulting deposition onto the MOF support. The here investigated series of isoreticular functionalized Pt@MOF composites comprises four materials based on Zn$_2$(DP-bdc)$_2$dabco (see Table 1).

After synthesis, all composites were washed, solvent exchanged, activated, and subsequently verified with respect to Pt content by atom absorption spectroscopy and photometry, and with respect to MOF composition and solvent removal by $^1$H NMR spectroscopy and TGA (see Supplementary Sections S1.3 and S4). PXRD

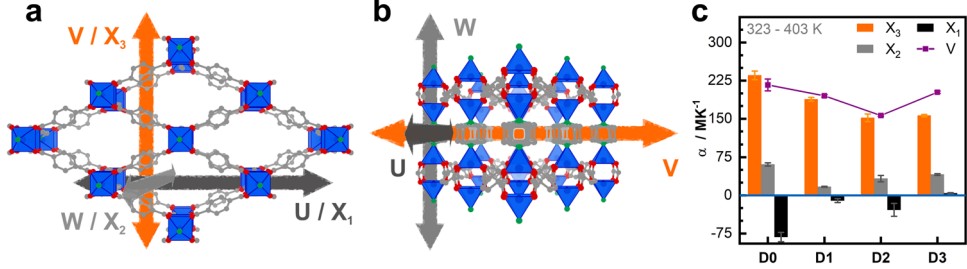

**Fig. 1 Crystal structure and thermal expansivity of Zn$_2$(DP-bdc)$_2$dabco.** View along (**a**) and perpendicular (**b**) to dabco pillars. Principal thermal expansion directions X$_1$, X$_2$, and X$_3$ approximately align to u, w, and v, respectively. **c** Thermal expansion coefficients (CTEs) for principal directions (columns) and net volume (connected points) of materials with and without Pt nanoparticle (**D#**: # wt.% Pt) are shown. See Supplementary Section S1.4 for structural formula of the linker, pillar, and paddlewheel unit. Grey, C; red, O; green, N; blue, Zn and coordination polyhedron; linker functionalizations and hydrogen atoms are not shown for clarity; error bars show standard deviation.

**Table 1 Pt content and thermal expansion coefficients $\alpha(X_{1-3})$ of Pt@MOF composites.**

| Pt@MOF | Pt [wt.%][a] | $\alpha(X_1)$[b] [MK$^{-1}$] | $\alpha(X_2)$[b] [MK$^{-1}$] | $\alpha(X_3)$[b] [MK$^{-1}$] |
|---|---|---|---|---|
| D0 | - | −82 | 61 | 235 |
| D1 | 1.4 | −10 | 17 | 188 |
| D2 | 1.9 | −29 | 32 | 152 |
| D3 | 3.1 | 4 | 41 | 156 |

[a]per atom absorption spectroscopy and photometry.
[b]more details in Supplementary Table 6.

confirms that all materials are isoreticular independent of the Pt content both in the as-synthesized state (**as**) directly after synthesis and washing, as well as in the activated state (**dry**) after solvent removal (see Supplementary Fig. 2). Materials crystallize in their large pore (lp) phase and are transitioned to their narrow pore (np) phase after solvent exchange and removal. This can be recognized by a shift of the 110 reflection and an intensity increase of the 001 reflection as reported in literature[37]. $Zn_2(DP\text{-}bdc)_2$dabco is further known to exhibit a reversible, $CO_2$ adsorption induced phase transition from the np to the lp phase[38]. Physisorption experiments reveal that this behavior is fully retained in all Pt@MOF composites as can be evidenced by their stepped $CO_2$ physisorption isotherms. NPs have a minor effect on accessible porosity and isotherm step shape (see Supplementary Fig. 3). The overall flexibility of this MOF system in response to solvents or adsorbates is briefly summarized in Supplementary Section S1.4. Thermogravimetric analysis (TGA) shows a Pt content correlated lowering of decomposition temperatures of up to 70 °C when comparing **D0** to **D3** under inert gas flow, while differential scanning calorimetry (DSC) confirms absence of any thermally induced first-order phase transitions irrespective of Pt loading (see Supplementary Fig. 4).

**Nanoparticle allocation.** The size, distribution and location of Pt nanoparticles were investigated by scanning transmission electron microscopy (STEM). The presence of Pt could be detected in all three samples (**D1-3**) by annular dark-field (ADF) STEM. The Pt nanoparticles could be localized based on the contrast difference which originates from a different electron density between Pt NPs and the MOF support (black background vs. grey to white Zn vs. bright white Pt). In all three samples we detected Pt NPs with approximately 1–3 nm in size with a relatively homogenous particle distribution across MOF crystals and limited aggregation. To further validate the presence of the Pt nanoparticles and to study their distribution, elemental mapping based on energy dispersive X-ray spectroscopy (EDS) was carried out. Both results are visualized in Fig. 2a and Supplementary Figs. 7–9. The recorded elemental distribution revealed a higher concentration of the Pt-element at the crystal edges which indicates presence of the Pt nanoparticles predominantly on the surface of the MOF crystals. To study the Pt nanoparticle distribution in more detail, electron tomography was conducted[39]. A series of STEM micrographs with incremental sample tilting steps were collected for composites **D1-3** (40–60 micrographs per sample; selection shown in Supplementary Fig. 10). This tilt-series was used to visualize the nanoparticles' position within the MOF crystals; see Supplementary Video 1 as example. Due to gradual sample decomposition under the electron beam during the series' acquisition and accompanied image quality consistency loss, only **D2** could be reconstructed by utilizing the TEMography™ software (Fig. 2b, see Supplementary Video 2); however, videos made from the recorded micrograph series of all three samples (**D1-3**)

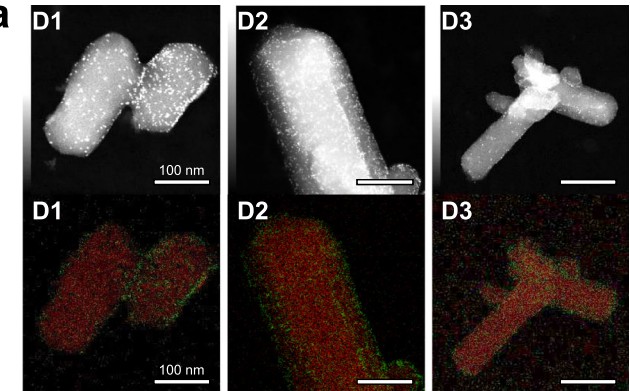

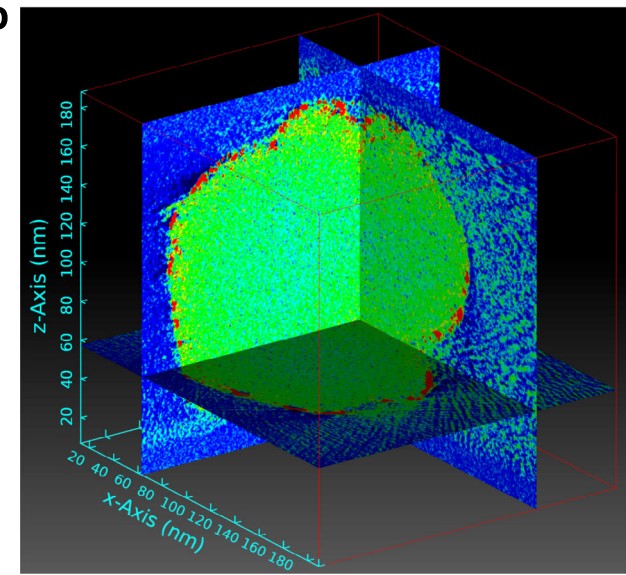

**Fig. 2 Electron microscopy and tomography analysis. a** ADF-STEM images (top row) showing Pt nanoparticles as bright dots distributed across the crystals and their corresponding EDS elemental mapping overlays (bottom row, contrast adjusted for visibility; red, Zn; green, Pt) revealing a higher concentration of the Pt-element along the edges of the MOF crystals. **b** A cut through a crystal of **D2** reconstructed from electron tomography, a video is available in the ESI. Red areas indicate high Z (Pt), while the MOF is represented by green color. Vacuum is colored blue.

clearly show a homogeneous distribution of uniform Pt nanoparticles deposited on the surface of the MOF crystals.

**Thermal expansion.** Thermal expansion behavior is investigated by variable temperature powder X-ray diffraction (VTPXRD). All composites were measured as **dry** samples in argon and therefore initially in their np state as shown in Fig. 1a, b and Supplementary Fig. 1. Each Pt@MOF composite was subjected to a heating and cooling cycle from 323 K to 513 K and back to 323 K. Diffraction patterns were collected every 20 K up to 443 K, every 10 K up to 513 K, and vice versa during cooling. The parent MOF $Zn_2(DP\text{-}bdc)_2$dabco (**D0**) reportedly exhibits significant thermal expansion and **D1**, **D2**, and **D3** show temperature-correlated shifts of reflection positions indicating similar responsiveness. An exemplary contour plot of VTPXRD data of **D1** is shown in Fig. 3 and plots of all other samples are given in the Supplementary Fig. 6. Most notable within the range of the composites' dominant reflections is a shift of the 110 peak from 2θ = 4.55– 4.40° during heating from 323 K to 443 K followed by a shift to 2θ = 4.35° by 513 K. Raw data shows the intensity of both 110 and 001 (at 2θ = 4.25°) reflections slightly increase with temperature.

Lattice parameters determined by Pawley profile fits and details of the outcomes are provided in tabulated form (Supplementary Table 2–5). In this work we focus to discuss volume changes and principal axis deformation instead of unit cell parameters for a better comparison with literature data. Accordingly, we transformed lattice parameters into principles axis by using PASCal[40], see Fig. 1 for a visualization of principal axis compared to unit cell directions and Supplementary Table 6 for a detailed overview. Fig. 4 shows the individual principal axis strains at each temperature step. All composites exhibit anisotropic thermal response and cell volume increases primarily due to elongation along the $X_3$ direction. The observed anisotropic TE is correlated to the anisotropic MOF topology: the pillared-layered $Zn_2(DP-bdc)_2dabco$ is built from paddlewheel metal-nodes each connecting four linkers to span two-dimensional square-lattice nets which are stacked by dabco pillars to form a 3D network (see Supplementary Fig. 1). This results in the so-called wine-rack motif and motion, i.e. a transformation between a lozenge-type and square-type structure, which has been reported to occur upon

external stimuli changes such as temperature[17] and pressure;[41] in the context of flexible MOFs, this phenomenon is defined as breathing[42]. This is possible because bdc-linker molecules and metal nodes can adapt to stress via bending of the carboxylate moiety and distortion of the paddlewheel coordination sphere to change linker-node-linker angles and effective linker length[43–45]. Projected onto this motif and structural transition, axis $X_3$ corresponds to the short direction [010] of the lozenge-lattice and axis $X_1$ is approximately projected as [100]/[101] along the long lozenge-lattice direction. When expanding from a lozenge- to a square-lattice, $X_3$ exhibits PTE and $X_1$ NTE. $X_2$ lies along [001], the stacked metal nodes and dabco pillars, and is mechanically stationary in this motion.

Upon heating, **D0** shows a 4.5% linear unit cell volume increase up to 493 K. This is congruent to the behavior of the isoreticular, literature-reported $Zn_2(BME-bdc)_2dabco$ (BME = 2,5-bismethoxyethoxy) which volume increases by 4.5% upon heating to 463 K[17]. Pt nanoparticles gradually decreases this to 3.9% (**D1** and **D2**) and 3.0% (**D3**). It is important to note, however, that for all materials two distinct temperature regions of thermal responsiveness can be observed; a low temperature region (up to approximately 403 K) where all materials display linear expansion behavior and a notable slope change in parameter evolution above this temperature for one or more unit cell parameters or principal directions, see Fig. 4. In the following, we will first elaborate on the linear TE region, then discuss the high-temperature behavior.

Within the lower temperature range, the most pronounced differences are observed from **D0** to **D1** indicating that changes in TE behavior are more related to Pt presence than amount. TE with the highest quality linear fits from 323 K to 403 K along each principal axis is summarized by the CTEs $\alpha(X_1)$, $\alpha(X_2)$, and $\alpha(X_3)$ in unit $MK^{-1}$, see Table 1 and Fig. 1c for an overview. **D0** exhibits anisotropic thermal expansion predominantly applying to $X_3$. Positive $\alpha(X_3)$ and negative $\alpha(X_1)$ correspond to the intuitively expected wine-rack opening motion which also aligns with the changes observed for the guest-induced phase transition of $Zn_2(DP-bdc)_2dabco$. This implies that this by guest adsorption in a single step triggered motion (see Supplementary Fig. 3) is also, but more gradually, accessible by temperature changes as second order phase transition. The addition of 1.4 wt.% Pt nanoparticles in **D1** leads to a partial inhibition of this wine-rack motion. In other words, for **D1** only PTE is present along the $X_3$ direction without concerted contraction of $X_1$. This effect persists for higher nanoparticle loadings in **D2** and **D3**. Furthermore, a relative reduction in $\alpha(X_3)$ of ca. 20% per step from **D0** > **D1** > **D2/D3** is observed. Interestingly, both absolute and relative changes in $\alpha(X_1)$ are higher than in $\alpha(X_3)$ when comparing **D0** to **D1** indicating

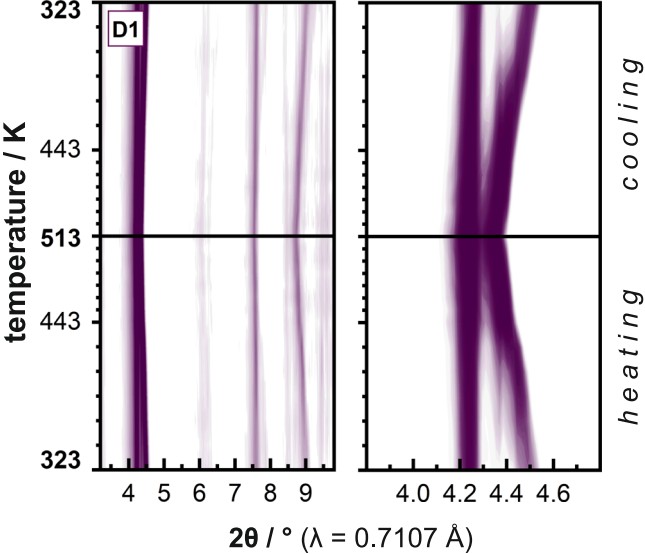

**Fig. 3 Contour plot of VTPXRD data of D1.** Darker purple corresponds to higher reflection intensity. Pattern in range of 2θ = 3.2–9.8° (left) and highlighted main reflections around 2θ = 4.4° (right). Each pattern is normalized to the single highest intensity signal of the complete pattern series. Contour plot and raw patterns for all materials are provided in Supplementary Section S5.

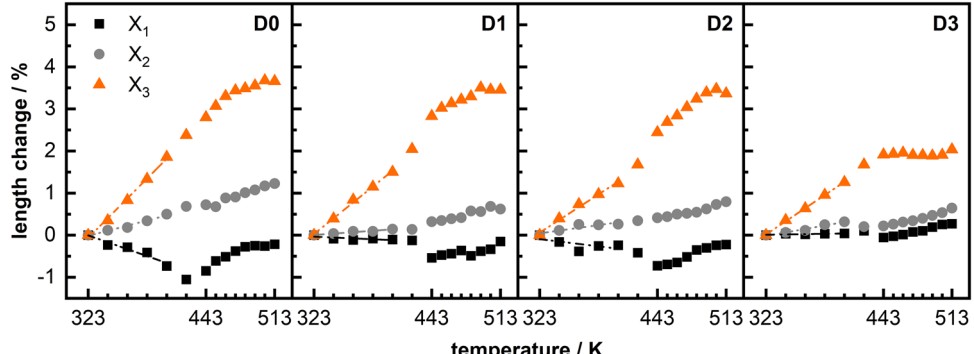

**Fig. 4 Thermal expansion along principal axes of D0-3.** Principal axes ($X_1$, $X_2$, and $X_3$) length change relative to values at 323 K calculated from lattice parameters using PASCal[40]. Fits as dotted lines for visualization of linear TE up to 403 K. Shown in all graphs is the heating branch; thermal expansion is fully reversible upon cooling (see Supplementary Section S6).

higher sensitivity of the contraction motion to nanoparticle presence than the expansion motion. For $\alpha(X_2)$ no clear correlation between nanoparticle loading and magnitude is obvious aside from a general slight reduction in TE in NP@MOF composites. Overall, when looking at the lower temperature behavior up to 403 K, deposition of Pt nanoparticles on the MOF inhibits the NTE observed along one of the three principal axes, especially in **D3**, while reducing PTE along the primary expansion axis by up to 35%.

In the pursuit to rationalize these observations, it is useful to treat Pt NPs as mesoscopic surface defects which are increasingly explored in flexible MOFs[46,47]. Recently Thompson et al.[48] investigated particle size-dependent flexibility of the pillared-layered MOF DUT-8 ($Zn_2ndc_2dabco$, $ndc^{2-}$=2,6-naphthalene dicarboxylate, DUT = Dresden University of Technology). DUT-8 is isostructural with $Zn_2(DP-bdc)_2dabco$ and large crystals of DUT-8 show an adsorbate-induced first-order phase transition that disappears for its fine-grained powder[45]. It was concluded that not specifically the particle size, but the amount of surface defects and affected nucleation barriers (of the phase transition) govern this difference and that surface modifications by modulators or capping agents offer the opportunity to manipulate this mode of flexibility. These considerations also apply to the composites studied in this work, albeit achievable effects are presumably damped by the fact that a gradual conformation shift and second order phase transition is observed instead of a first order transformation.

An additional perspective on the effect of surface mounted NP on the TE properties is offered when correlating effects of NPs to classical mechanics, effectively treating NPs as a constraint for the network motion of $Zn_2(DP-bdc)_2dabco$. The topology elaboration above and visualization in Fig. 5 imply that nanoparticles that are surface mounted on the plane spanned by $X_1$ and $X_3$ are expected to have a pronounced impact on the wine-rack type motion, effectively blocking any network motion that comes from bond angle, conformation, and geometry changes. While $Zn_2(DP-bdc)_2dabco$ is no ideal joint-and-bar assembly, our experimental data, especially the absence of temperature-dependent peak broadening in the PXRD pattern, suggest that this effect is large enough to translate to the bulk. Complete rigidification, however, is not expected due to the anisotropic shape of the MOF crystallites which, based on reported behavior of the topologically comparable DUT-8 needle-like crystals[48], grow along the pillar/dabco direction. This presents an excess of side facets, in turn only a few NPs exhibit direct influence on the wine-rack motion, and is the surmised reason why a gradual Pt loading-dependent change in TE behavior is observable in this system, as opposed to complete rigidification with NPs present.

Looking at higher temperatures, deviations from the linear lower-temperature behavior are observable. We attribute this to the gradual second order phase transition linked to the wine-rack motion of the framework becoming more prominent at higher temperatures. The related slope change in principal axis strain progression is most visible in $X_1$ for **D0**, **D1**, and **D2** and in $X_3$ for **D3**. In **D0**, after 423 K, the expansion direction of $X_1$ is inverted and the expansion rate of $X_3$ decreases; $X_2$ seems unaffected. **D1** and **D2** mirror this behavior slightly delayed at 443 K. It is less defined in $X_1$ because the nanoparticles seem to rigidify especially this contractive deformation as was observed at lower temperatures. Notably, PTE in **D3** changes drastically at 423 K. Expansion of the $X_3$ axis is halted completely until 513 K with a narrow plateau around 443 K close to ZTE. Above 453 K, the temperature-induced strain results again in gradual but reduced TE, however, by minor elongation along $X_1$ and $X_2$ as new dominant expansion axes. Overall, when looking at the higher temperature behavior above 443 K, the small step in Pt loading from **D2** to **D3** proves most influential. This suggests that increased nanoparticle loading might in general have higher TE impact at higher temperatures and deposition of otherwise innocent nanoparticles might be of interest to reduce strain exerted onto porous material by their NTE or PTE.

## Conclusions

In conclusion, we have developed a one-pot, capping-agent free synthesis route for Pt@ $Zn_2(DP-bdc)_2dabco$ composites with precise control over homogeneity and nanoparticle surface-loading which enables the study of surface-mounted nanoparticle formation on the TE properties of the MOF support. The surface-mounted platinum nanoparticles in the size regime <3 nm display distinctly different influence on the anisotropic (net positive) thermal expansion of the $Zn_2(DP-bdc)_2dabco$ support depending on the temperature range observed. First, at temperatures up to 403 K, gradually increasing amounts of Pt NPs proportionally decrease expansion along the main expansion axis by up to 35%, while the introduction of low nanoparticle amounts (1.4 wt.%) significantly inhibits contraction along the secondary expansion axes. Up to 403 K, all investigated composites show reduced but retained net PTE. Second, at temperatures above 403 K, the highest nanoparticle loading (3.1 wt.%) resulted in a large change in TE behavior. Expansion along the main axis is halted from 423 K to 513 K with a narrow plateau close to ZTE around 443 K after which only minor net PTE is observed due to strain along the other axes. We conclude from this that at low nanoparticle loadings and temperatures the TE of NP@MOF composites can be approximated by the MOF support. It is implicated that composite materials with higher NP loading exhibit superior TE properties compared to the pure MOF because their thermal stress is reduced, especially at higher temperatures. It is further intriguing that specifically NTE along a principal direction was inhibited most by the nanoparticles. If, hypothetically, their influence is more pronounced on NTE than PTE, this might open another parameter space for MOF fine-tuning given how prominent NTE is within this material class.

While this study focused on a single model material to highlight temperature and loading dependencies, it is reasonable to assume other NP@MOF composites undergo similar changes to their thermal responsiveness. TE in such composites expectedly correlates to the interplay of nanoparticle loading, metal (oxide), size, and capping agents - all of which are firmly tied to the synthetic challenges of precise NP bulk-incorporation and surface-deposition - as well as MOF support topology, particle size, defect amount, and more. Future studies are envisioned to expand on the phenomenological investigation and, in close

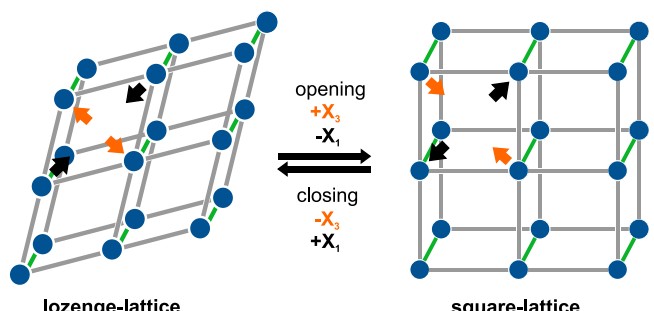

**Fig. 5 Thermally induced framework motion.** Anisotropic structural transition between lozenge-lattice (left) and square-lattice (right) of a wine-rack motif. Changes along principal axes $X_3$ (orange) and $X_1$ (black) are indicated by directed arrows. Structure represents paddlewheel nodes (blue), linker struts (grey) and pillar struts (green).

collaboration with molecular dynamic simulations, to unravel detailed mechanisms of this complex interplay. Understanding this will play an important role for increasing longevity and performance in applications with repeating heating or cooling cycles which, incidentally, is very likely to be the case in the most prominent uses of these composites as adsorbents, sensors, or catalysts.

## Methods

**Material synthesis**. All chemicals were purchased from commercial suppliers (Sigma-Aldrich, Fluka, Alfa Aesar, and others) and were used without further purification. Ethanol for solvent exchange was purchased as technical grade and redistilled prior to use. The linker was synthesized via Williamson Etherification according to literature known procedures, albeit slightly optimized[38]. All Pt@Zn$_2$(DP-bdc)$_2$dabco composites were synthesized via a concerted one-pot crystallization of MOF and formation of Pt nanoparticles without any capping agents[34]. This method combines the solvothermal MOF synthesis in dimethyl-formamide (DMF), as known from literature[17], with a solvent mediated Pt$^{2+}$ reduction. Multiple washing and solvent exchange steps were conducted prior to solvent removal *in vacuo*. See Supplementary Methods for details.

**Analytical methods**. Liquid state NMR spectra were measured on a Bruker Ultrashield DRX400 spectrometer at ambient temperature (see Supplementary Section S1.1). MOF samples were digested and subsequently measured in DMSO-d$_6$ with DCl, organic linkers were dissolved and measured in DMSO-d$_6$. CHNS contents were determined via combustion analysis. Adsorption measurements were carried out on a 3Flex Physisorption Instrument by Micromeritics Instrument Corp. Samples were activated at 80 °C for 5 h under dynamic vacuum using a SmartVacPrep by Micromeritics Instrument Corp. to ensure absence of unwanted adsorbates. The adsorbent mass was then recorded, generally in the range of 50–70 mg (for more method details see Supplementary Section S3). All adsorption isotherms are accessible online free of charge as adsorption information files to counter issues with post-publishing adsorption data extraction and facilitate machine learning (see Supplementary Data)[49]. Thermogravimetric analysis coupled with differential scanning calorimetry (TGA-DSC) was conducted on a Netzsch TG-DSC STA 449 F5 in a temperature range from 25 °C to 800 °C with a heating rate of 10 K min$^{-1}$ under argon flow (see Supplementary Fig. 4). Powder X-ray diffraction (PXRD) patterns of the as-synthesized (**as**) and re-solvated samples were measured on a Rigaku Benchtop MiniFlex 600-C (X-ray Cu Kα radiation, λ = 1.5406 Å). Activated (**dry**) samples and samples after physisorption measurements were measured in Debeye-Scherrer geometry on a PANalytical Empyrean (X-ray Cu Kα radiation, λ = 1.5406 Å) (for further details see Supplementary Section S2). Variable temperature PXRD (VTPXRD) measurements were done using a STOE INSITU HT2 Furnace on a STOE STADI P Dual (X-ray Mo Kα radiation, λ = 0.7107 Å). **Dry** samples were filled in borosilicate glass capillaries in a glovebox under argon atmosphere. Sealed capillaries were inserted into an open 1 mm diameter quartz capillary and measured in Debeye-Scherrer geometry from 323 K to 513 K to 323 K (see Supplementary Section S5). Scanning transmission electron microscopy (STEM) with energy dispersive X-ray spectroscopy (EDS) elemental mappings and electron tomography were recorded with a JEM-ARM200F NEOARM microscope from JEOL (Germany) GmbH with a cold FEG electron source operated at 200 kV. Samples were prepared by depositing a drop of the solid dispersed in ethanol onto carbon-coated copper grids (200 mesh) and dried in air (see Supplementary Section S8).

## Data availability

The primary data that support the findings of this study are available within the supplementary information files and from either corresponding author upon request.

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

## Author contributions

J.B. performed material synthesis, analysis, data processing and manuscript writing. A.-S. D. contributed to material prototyping, analysis, and manuscript writing. A.U. and H.B. performed all electron microscopy, its visualization, and contributed to manuscript writing. R.A.F. provided the motivation of this work and contributed to manuscript writing. G.K. provided the motivation of this work and contributed to data processing and manuscript writing.

## Funding

This work was funded by the DFG via the research unit FOR 2433 (MOF Switches). Open Access funding enabled and organized by Projekt DEAL.

## Competing interests

The authors declare no competing interests.
