## [Peer Review File · Communications Chemistry]

Reviewers' comments:

Reviewer #1 (Remarks to the Author):

The manuscript emphasizes the importance of the control over thermal expansion coefficients of nanoparticle-MOF composites, which are considered as promising materials for application in catalysis. The authors demonstrated that TE coefficients of NP@MOF composites are strongly dependent on the loading and can even be switched from negative to positive values for defined principal directions. All in all the manuscript presents a new and original concept for control over TE, which can be interesting for a broad readership involving such disciplines as inorganic chemistry, physical chemistry and catalysis. In principle, the manuscript can be considered for publication, however, some additional justification may be required. Although the analysis of the data is conducted and presented at a high level, the discussion of the possible reasons for the difference of TE along the principal directions is not present. Since Pt nanoparticles are homogeneously distributed on the crystallite surfaces, authors may check for some correlations between the change in TE parameters along the principal directions and the surface of the corresponding crystal face.

Few minor points to be considered:

- 1) It may be useful to insert the structure or graphical formula of H₂DP-bdc.
- 2) In Supplementary figure 1 PXRDs are shown from 6 to 24° 2 θ . Despite of low mass.% of Pt, it may be reasonable to check PXRD at higher angles to check if some reflections of Pt nanoparticles can be observed.
- 3) Additional explanation of structural flexibility of Zn₂(DP-bdc)₂dabco should be added to ESI. This may increase the understanding of the phenomena for the broader readership.

Reviewer #2 (Remarks to the Author):

Berger et al. are here explore the effect of nanoparticles on the thermal expansion of a prototypical MOF, Zn bdc dabco. They show that deposition of nanoparticles on the surface of the MOF reduce the magnitude of thermal expansion. MOF composites are of increasing importance for a variety of applications, and the Pt@MOF system is well chosen for its relevance to catalysis. The paper contains high quality data throughout and all compounds are very well characterised. In particular the XRD data are high quality and treated with due care for the range of complex issues that occur in this material. This is an interesting and important study and should be published, subject to some minor issues below.

1. Can anything about changes in the CO₂ induced gate opening due to Pt be said?
2. " During solvothermal MOF synthesis, DMF oxidizes with water originating from trace amounts in the DMF and the hydrated metal salt to its carbamic acid derivative, which reduces Pt²⁺ and facilitates nanoparticle formation."

An oxidised species would not be the reducing agent.

3. Figure 3 is nice, however peaks should be normalised to the same intensity for every pattern (i.e. divide by the maximum) as flux variations are likely to be smaller than the sample dependent peak intensity variation (unless there is some other issue). Contrast around Bragg peaks for figure 3 RHS should be improved (i.e. change the colorbar scaling).

4. "D0 shows a 4.5 % linear unit cell volume increase up to 493 K which is in very good agreement to literature reports"

The authors should quote the literature values in text.

5. "Precise modulation of reaction temperature, time and precursor concentration allows for control over Pt nanoparticle formation kinetics and resulting deposition onto the MOF support."

The authors don't really describe anything precise. Please avoid puffery, it's unnecessary and obscures meaning.

6. Some typos/formatting points:

"comprises of four materials"

comprises doesn't take an a preposition: should be "comprises four materials"

"This is can be recognized"

(110) reflection

Pedantic but reflections aren't planes and so don't need brackets round the miller indices. If they were {} are needed because it is a powder pattern. True for all peaks.

Figure S1 is a little pixellated - as this is the core structural characterisation it would be nice to have it big and clear.

Reviewer #3 (Remarks to the Author):

The authors, Berger et al., report on the thermal expansion (TE) properties of platinum nanoparticle loaded zinc MOFs and the impact of nanoparticle loading on the change in coefficients of thermal expansion. Although catalytic properties and applications of nanoparticle-MOF composites are widely reported, it is surprising that the fundamental thermal properties are not more investigated. Using reported procedures, the authors synthesize Pt@MOF and perform variable temperature powder X-ray diffraction and observed that loading of platinum nanoparticles reduces the thermal expansion of the MOF and identify the phase changes during thermal expansion that cause the changes in thermal expansion. The authors also evaluate the impact of amount of nanoparticles deposited, on the thermal expansion of Pt@MOF and identify that temperature ranges and the crystallographic axes where the change in thermal expansion is significant.

The authors claim that the deposited NPs disrupts the thermal induced wine-rack motion of the MOF resulting in a decrease in the overall thermal expansion. My overall assessment is that while the findings are novel and interesting, the explanations advanced are not convincing. Basically, how can nanoparticles on the surface of a material change its bulk thermal expansion? Without a robust explanation the report is phenomenological.

Response to reviewer comments

Reviewer #1 (Remarks to the Author):

The manuscript emphasizes the importance of the control over thermal expansion coefficients of nanoparticle-MOF composites, which are considered as promising materials for application in catalysis. The authors demonstrated that TE coefficients of NP@MOF composites are strongly dependent on the loading and can even be switched from negative to positive values for defined principal directions. All in all the manuscript presents a new and original concept for control over TE, which can be interesting for a broad readership involving such disciplines as inorganic chemistry, physical chemistry and catalysis. In principle, the manuscript can be considered for publication, however, some additional justification may be required.

Although the analysis of the data is conducted and presented at a high level, the discussion of the possible reasons for the difference of TE along the principal directions is not present. Since Pt nanoparticles are homogeneously distributed on the crystallite surfaces, authors may check for some correlations between the change in TE parameters along the principal directions and the surface of the corresponding crystal face.

We appreciate the referee's positive feedback, interest in the work and constructive criticism which helped us to improve our work. In the revised version we expanded the description of the structural flexibility of the parent MOF which enables a more detailed discussion of how surface-mounted nanoparticles impact thermal expansion of the MOF. In short, we rationalize the impact of surface mounted nanoparticles via surface modifications and topological considerations, which both put mechanical restrictions on the bulk properties, an effect that was previously discussed for the particle-size dependent flexibility of DUT-8, see page 11f and 14f. A detailed, atomistic study via molecular dynamic simulations is on the way but beyond this experimental proof-of-principle study.

Few minor points to be considered:

- 1) It may be useful to insert the structure or graphical formula of H2DP-bdc.

The structure of the linker, pillar and a paddlewheel building unit has been added to the supplementary information (Supplementary Section S1.4) and is referenced in the main text (see page 4 and caption Figure 1).

- 2) In Supplementary figure 1 PXRDs are shown from 6 to 24° 2 θ . Despite of low mass.% of Pt, it may be reasonable to check PXRD at higher angles to check if some reflections of Pt nanoparticles can be observed.

This is an excellent point. In the revised manuscript we have increased the 2 θ range of the PXRD pattern to 6-50°, which includes angles where reflections of Pt can be expected (I(111) at approx.. 40° and I(200) at approx.. 47°). No Pt reflections can be observed which is expected due to the very

low loading of the nanoparticles and significant peak broadening coming from diffraction domains in the nano regime.

- 3) Additional explanation of structural flexibility of $\text{Zn}_2(\text{DP-bdc})_2\text{dabco}$ should be added to ESI. This may increase the understanding of the phenomena for the broader readership.

Thank you for bringing up this important point. In the revised version we added a more elaborate discussion on structural flexibility of the parent $\text{Zn}_2(\text{DP-bdc})_2\text{dabco}$ material, which serves as a starting point to understand the impact of nanoparticles on its thermal expansion behavior, see Supplementary Section 1.4 and page 11f.

Reviewer #2 (Remarks to the Author):

Berger et al. are here explore the effect of nanoparticles on the thermal expansion of a prototypical MOF, Zn bdc dabco. They show that deposition of nanoparticles on the surface of the MOF reduce the magnitude of thermal expansion. MOF composites are of increasing importance for a variety of applications, and the Pt@MOF system is well chosen for its relevance to catalysis. The paper contains high quality data throughout and all compounds are very well characterized. In particular the XRD data are high quality and treated with due care for the range of complex issues that occur in this material. This is an interesting and important study and should be published, subject to some minor issues below.

We are grateful for the constructive and positive review.

1. Can anything about changes in the CO₂ induced gate opening due to Pt be said?

This is a very intriguing question. Our CO₂ isotherms (Supplementary Figure 3) suggest that the Pt NPs slightly rigidify the underlying framework which results in a flatter step/curve during adsorption, although not an overall shift of the gate opening/closing pressure. The impact on the gate-opening pressure/general switchability (with various stimuli) of such and similar Pt@flexible MOF composites, however, diverges from the TE focus set in this work and is subject of ongoing different studies which we hope to share in the near future.

2. "During solvothermal MOF synthesis, DMF oxidizes with water originating from trace amounts in the DMF and the hydrated metal salt to its carbamic acid derivative, which reduces Pt²⁺ and facilitates nanoparticle formation." An oxidized species would not be the reducing agent.

Thank you for highlighting this confusion. We rephrased the section to emphasize the oxidation of DMF to carbamic acid occurs in tandem with the reduction of Pt²⁺ to Pt⁰ with DMF as reducing agent and carbamic acid as product (page 6).

3. Figure 3 is nice, however peaks should be normalised to the same intensity for every pattern (i.e. divide by the maximum) as flux variations are likely to be smaller than the sample dependent peak intensity variation (unless there is some other issue). Contrast around Bragg peaks for figure 3 RHS should be improved (i.e. change the colorbar scaling).

Thank you for this feedback. In the revised manuscript the colorbar scaling is improved for better visualization. Intensity normalization of peak intensities was already performed as suggested (single value of the maximum intensity observed across all patterns as reference for all individual patterns) and a comment has been added to the caption for clarification.

4. "D0 shows a 4.5 % linear unit cell volume increase up to 493 K which is in very good agreement to literature reports". The authors should quote the literature values in text.

This has been done, see page 13 (top).

5. "Precise modulation of reaction temperature, time and precursor concentration allows for control over Pt nanoparticle formation kinetics and resulting deposition onto the MOF support." The authors don't really describe anything precise. Please avoid puffery, it's unnecessary and obscures meaning.

Thank you for this remark. We wholeheartedly agree, removed this hyperbolic descriptor, and reworded another similar instance as well.

6. Some typos/formatting points:

Thank you for the following corrections. All have been addressed.

- a. "comprises of four materials" comprises doesn't take an a preposition: should be "comprises four materials"
 - b. "This is can be recognized"
 - c. Pedantic but reflections aren't planes and so don't need brackets round the miller indices. If they were {} are needed because it is a powder pattern. True for all peaks.
7. Figure S1 is a little pixellated - as this is the core structural characterisation it would be nice to have it big and clear.

Thank you for the remark. We both added a wider version of said figure as response to another reviewer comment, improved formatting and resolution, and provided a more zoomed-in addition to highlight the pattern changes during activation at lower angles.

Reviewer #3 (Remarks to the Author):

The authors, Berger et al., report on the thermal expansion (TE) properties of platinum nanoparticle loaded zinc MOFs and the impact of nanoparticle loading on the change in coefficients of thermal expansion. Although catalytic properties and applications of nanoparticle-MOF composites are widely reported, it is surprising that the fundamental thermal properties are not more investigated. Using reported procedures, the authors synthesize Pt@MOF and perform variable temperature powder X-ray diffraction and observed that loading of platinum nanoparticles reduces the thermal expansion of the MOF and identify the phase changes during thermal expansion that cause the changes in thermal expansion. The authors also evaluate the impact of amount of nanoparticles deposited, on the thermal expansion of Pt@MOF and identify that temperature ranges and the crystallographic axes where the change in thermal expansion is significant.

The authors claim that the deposited NPs disrupts the thermal induced wine-rack motion of the MOF resulting in a decrease in the overall thermal expansion. My overall assessment is that while the findings are novel and interesting, the explanations advanced are not convincing. Basically, how can nanoparticles on the surface of a material change its bulk thermal expansion? Without a robust explanation the report is phenomenological.

Thank you for the constructive review and interest in our work. In the revised version we have added a more elaborate discussion on the general flexibility of the MOF as a function of temperature changes (page 11f, Figure 5 and Supplementary Section 1.4), and more specifically address the impact of surface-mounted nanoparticles (page 14f). In short:

- By treating NPs as surface defects, studies on (surface) defect directed bulk properties of flexible MOFs can be referenced. Recently Thompson et al. (new Ref. 48) has investigated the particle size-dependent flexibility of a structurally related pillared-layered MOF DUT-8 (Zn_2ndc_2dabco), concluding that the amount of surface defects is the decisive factor for flexibility rather than particle size. This reasoning applies to herein studied composites due to the shared topology, nevertheless, achievable effects are expected to be damped by the fact that a gradual conformation shift is observed in the MOF $Zn_2DP-bdc_2dabco$ instead of a first order transformation in DUT-8.
- An alternative perspective that enables to qualitatively understand the anisotropic effects of NPs on the motion relies on classical mechanics, where surface mounted NPs act as constraints for the wine-rack type structural flexibility. Complete rigidification, however, is arguably not observed due to the anisotropic particle size and structure of the MOF, and only NP deposited on the (X_1, X_3) plane are expected to impact the wine-rack motion.

REVIEWERS' COMMENTS:

Reviewer #1 (Remarks to the Author):

The authors considered all comments and suggestions of the reviewers and improved the manuscript by discussing these points in the main text and in ESI. In its current form, the manuscript can be recommended for publication.

Reviewer #2 (Remarks to the Author):

Changes are positive should be accepted.